# Oral Anticoagulant Therapy—When Art Meets Science

**DOI:** 10.3390/jcm8101747

**Published:** 2019-10-21

**Authors:** Patricia Lorena Cîmpan, Romeo Ioan Chira, Mihaela Mocan, Florin Petru Anton, Anca Daniela Farcaş

**Affiliations:** 1Department of Cardiology, Heart Institute, 40001 Cluj Napoca, Romania; pcimpan@gmail.com; 2Internal Medicine Department, Iuliu Hatieganu University of Medicine and Pharmacy, 400012 Cluj-Napoca, Romania; mihaela.mocan@gmail.com (M.M.); ancafarcas@yahoo.com (A.D.F.); 3Emergency Clinical County Hospital, 40006 Cluj Napoca, Romania

**Keywords:** anticoagulant, *CYP2C9*, *VKORC1*

## Abstract

Anticoagulant treatment is extremely important and frequently encountered in the therapy of various cardiovascular diseases. Vitamin K antagonists (VKA) are in use for the prevention and treatment of arterial and venous thromboembolism, despite the introduction of new direct-acting oral anticoagulants (NOAC). The VKA still have the clear recommendation in patients with a mechanical prosthetic heart valve replacement or moderate to severe mitral stenosis of the rheumatic origin, in deep vein thrombosis associated with congenital thrombophilia, and in cases where NOAC are prohibited by social condition (financial reason) or by comorbidities (extreme weight, severe renal or liver disease). VKA dosing required to reach the targeted therapeutic range varies largely between patients (inter-individual variability). This inter-individual variability depends on multiple environmental factors such as age, mass, diet, etc. but it is also influenced by genetic determinism. About 30 genes implicated in the metabolism coumarins derivatives were identified, the most important being *CYP2C9* and *VKORC*, each with several polymorphisms. Herein, we review the data regarding genetic alterations in general and specific populations, highlight the diagnosis options in particular cases presenting with genetic alteration causing higher sensitivity and/or resistance to VKA therapy and underline the utility of NOAC in solving such rare and difficult problems.

## 1. Introduction

Anticoagulant treatment is extremely important and frequently encountered in the therapy of various cardiovascular diseases. In daily practice, the achievement and maintenance of efficient anticoagulation with vitamin K antagonists (VKA) is difficult because of the narrow target range with the adverse effects of overdosing associated with hemorrhagic events and suboptimal dosing responsible for thromboembolic events. The risk of hemorrhagic events associated with a higher level of International Normalized Ratio (INR) is greater, with 10% to 17% occurring during the first weeks of treatment. Most of the major bleedings take place after the initiation of the therapy, with the hemorrhagic risk being 10 times higher in the first month than in the twelfth month of anticoagulant treatment [1]. On the other hand, some studies showed that suboptimal anticoagulation was associated with poor clinical outcomes, a higher rate of hospitalization, and increased risk for ischemic stroke and other thromboembolic events [2]. 

## 2. General Consideration on VKA

VKA has been the most prescribed oral anticoagulants for the prevention and treatment of arterial and venous thromboembolism in the last 60 years, until the entrance on the market of four new direct oral anticoagulants (NOAC): dabigatran, rivaroxaban, apixaban and edoxaban [3]. At the moment, based on newly developed European and American guidelines, VKA treatment remained the first line of therapy in certain groups of patients, such as those with a mechanical prosthetic heart valve replacement or atrial fibrillation in moderate to severe mitral stenosis of a rheumatic origin [4]. VKA treatment may be a valid option in patients with atrial fibrillation and biological valves or after valve repair, with these patients being in a grey area [4]. 

The same grey area is applicable to the use of oral anticoagulation therapy in patients undergoing transcatheter aortic implantation (TAVI). The extension of TAVI to patients at low operatory risk will increase the number of people treated with this technique, thus the importance of understanding when and which anticoagulant agent needs to be used. A further and increasing concern regards the risk of valve thrombosis after TAVI with most cases occurring within one year of the procedure, with a median onset of three to six months. The lack of post-TAVI warfarin treatment was an independent predictor of valve thrombosis [5,6]. Small population trials fail to provide significant net clinical benefits of dual antiaggregant therapy over single antiplatelet therapy [7]. The 2017 updated version of AHA/ACC guidelines also recommend the possibility of an anticoagulation-based strategy with VKA to achieve an INR of 2.5 for at least three months after TAVI in patients at a low bleeding risk [8]. 2017 ESC/EACTS Guidelines for the management of valvular diseases recommend, despite the lack of evidence, the use of NOAC in patients who have atrial fibrillation associated with an aortic bioprosthesis >3 months after implantation [9]. There are some trials ongoing and aiming to compare NOAC to warfarin in TAVI patients with underlying atrial fibrillation, or another reason to receive oral anticoagulation. These studies may very soon solve this issue [8].

VKA treatment is recommended in patients with antiphospholipid syndrome (APS), which is a heterogeneous group, with different clinical phenotypes and risk profiles, prone to recurrent thrombosis [10]. For patients with venous or arterial thrombosis secondary to APS long-term anticoagulant therapy with a VKA is recommended (target INR ranges between 2 to 3) [11]. Higher-intensity anticoagulation (INR 3–4) did not improve outcomes in venous thrombosis but is preferred for arterial thrombosis in some centers [12]. Even though trials of NOAC for patients with APS are ongoing, for now, there is insufficient data to use them on a large scale in this patient population [13]. Another group of patients who benefit from VKA treatment is those suffering from recurrent thrombosis or thrombosis in atypical sites associated with major congenital thrombophilia [14]. But even in these particular cases, NOAC appear to be efficient and safe [15]. At last, VKA treatment might be the patient’s choice on different grounds such as costs. The physician should also have in mind the patient’s adherence to treatment when choosing the anticoagulant type. Surprisingly, some studies showed that discontinuation of anticoagulant treatment at one year was higher in patients with NOAC than in those with VKA [16]. Overall, factors such as patient preference, patient compliance, the type of previous thrombotic events (provoked or unprovoked, arterial or venous), the lability of INR values, and the cost and quality of life should be taken into consideration when choosing the anticoagulant therapy. 

Worldwide, the most frequently prescribed coumarin oral anticoagulant drugs are warfarin, acenocoumarol and phenprocoumon [17]. Warfarin, the first oral anticoagulation drug, has a fascinating story dating back in 1921, during the cattle fever in Alberta, and North Wisconsin, in the USA, when Frank Schofield and Lee Roderick, local veterinary surgeons, first diagnosed “sweet-clover disease” [17]. It took Karl Link six years to isolate 3,3′-methylenebis (4-hydroxycoumarin) or dicoumarol, the chemical responsible for sweet-clover disease, from the liver of dead cows [18]. “Link proposed that coumarin derivative should be used as a rodenticide. Among various modified forms of dicoumarol, compound 42 was found to be more effective and was named as WARFARIN—named from Wisconsin Alumni Research Foundation and the “arin” from coumarin” [19]. Warfarin transitioned into clinical use under the trade name Coumadin and was approved for use in humans in 1954 and became the most used oral anticoagulant worldwide [20]. Currently, its use is decreasing in favor of NOAC but it still remains one of the greatest discoveries of the last century.

All coumarinic derivatives are 4-hydroxy-coumarines and are formulated as a racemic mix of 50% S-enantiomer and 50% R-enantiomer. Even though the action mechanism of the anticoagulation process is similar, there are some important differences in the pharmacokinetics of the three coumarin drugs.

First of all, all coumarins, except for S-acenocoumarol, are absorbed from the digestive tract having an almost complete oral bioavailability. S-acenocoumarol has a more intense metabolism during the first passage, reaching the maximum plasmatic concentration during the first few hours after intake, with almost 98%–99% of coumarin being bound to plasmatic albumin [21].

Secondly, VKAs are metabolized in the liver by hydroxylation involving cytochrome (CYP) P450 enzymes. S-warfarin (the most active form) is mainly metabolized by *CYP2C9* while R-warfarin is metabolized by several other CYP isoforms [22]. *CYP2C9* is also the main enzyme involved in the metabolism of both enantiomers of acenocoumarol and plays a less important part in phenprocoumon metabolism (where *CYP3A4* is mainly involved) [23].

Thirdly, of the three VKAs, phenprocoumon has the longest plasma half-life—110–130 h [24], followed by 35–58 h for R-warfarin, 24–33 h for S-warfarina, while acenocoumarol has the shortest half-life. Although the S enantiomer is more active, it has a very short half-life and the anticoagulation depends mostly on the R enantiomer (with a half-life of 6.6 h) [25].

The VKA dose needed to reach effective anticoagulation varies on an individual basis, up to 20-fold for warfarin [26] and up to 5-fold for acenocoumarol [27], therefore the management of anticoagulation poses great difficulties due to the inter-individual variability of the response. The inter-individual variability depends on environmental factors—age, weight, diet, vitamin K intake, concomitant medication, sex, liver diseases, congestive heart failure, serum albumin levels, cytochrome P450 polymorphism, alcohol intake, and patient compliance [28], but has also a genetic determinism [29]. 

At this point, Factor VII (FVII) deficiency (inherited or acquired) is worth mentioning, as it interferes with coagulation that interferes with VKA control. While the inherited FVII deficiency is an autosomal recessive bleeding disorder, the acquired forms are sporadic and frequently associated with autoimmune disorders and neoplastic conditions [30]. The diagnosis hallmark of FVII deficiency is an elevated INR in the absence of liver disease and a normal activated partial thromboplastin time (aPTT) [31]. Thus, baseline INR is elevated precluding monitoring with usual INR goals, and there is limited data regarding anticoagulation therapy in such rare cases. Alternative solutions to INR monitoring were proposed in short case reports. The authors recommended monitoring factor II (FII) and factor X (FX) which are VK dependent [32]. Their activity correlates linearly with INR and thus, obtained INR ranges can be used for VKA dosing [33].

As for the genetic mutation affecting the response to VKA therapy, more than 30 genes are involved in VKA activity and metabolism; the most important and most extensively studied are *CYP2C9* (the gene responsible for the enzyme mainly involved in the metabolism of S-warfarin) [22,34] and *VKORC1* (the gene encoding the target enzyme of the medication—vitamin K epoxide reductase) [35,36,37] (Figure 1). 

Herein, we review the data regarding genetic alterations in general and specific population, highlight the diagnosis options in particular cases presenting with genetic alteration causing higher sensitivity and/or resistance to VKA therapy and underline the utility of NOAC in solving such rare and difficult problems.

## 3. Genetic Determinism of the Response to VKA 

### 3.1. What Does CYP2C9 Means?

The genes encoding *CYP2C9* have several polymorphisms (with a variable incidence in various populations) that cause a decrease in the enzyme activity and a variable effect on VKAs metabolism. There are many mutant alleles of *CYP2C–CYP2C9*1* (wild allele—considered the most common variant), *CYP2C9*2* (Arginine>Cysteine at codon 144), *CYP2C9*3* (Isoleucine>Leucine at codon 359), *CYP2C9*4* (less frequently found, Isoleucine>Threonine at codon 359), *CYP2C9*5* (Aspartic acid>Glutamic acid at codon 360) etc. The most frequent and most studied are the *CYP2C9*2* and *CYP2C9*3* alleles. Over 30% of the European and Caucasian population has one or both alleles mutant, meanwhile they are extremely rare in Asian and African-American populations, where over 95% of people have the wild allele. These two variants decrease the warfarin metabolism by 30%–50% (*CYP2C9*2*) and by up to 90% (*CYP2C9*3*), whereas their association further decreases the metabolic efficiency of *CYP2C9* enzyme [29,40,41]. Some of the mutations are responsible for decreasing the VKA metabolism and predisposing to major bleeding events under VKA. This phenomenon is called high sensitivity to VKA [42]. Moreover, compared to the *CYP2C9*1/*1* genotype, the *CYP2C9*1/*2, *1/*3* și **3/*3* genotypes decrease the S-warfarin clearance with approximately 40%, 60% and 90%, respectively [43,44], that accounts for 10–15% of dose variance for warfarin and 5% for acenocoumarol in Caucasians [27] but significantly less in African-Americans, due to their lower incidence in this population [45].

Therefore, in order to predict the AVKs dose, several algorithms have been devised, that associate clinical and demographic factors to *CYP2C9*2* and **3* polymorphisms. This association has increased the predictive ability for the AVKs dose, depending on the allele incidence in the studied population [46], leading to an increase of 4.7% [46], 11.7% [47] and 14% [48], respectively for Romanian [46], Spanish [49] and French [48], respectively (for acenocoumarol). In the case of warfarin, the addition of *CYP2C*2* and **3* polymorphisms to age and weight [50] or age, sex, ethnicity, and concomitant the number of medications [51] has explained an additional 17.5% [50] and 12% [51] of the dose variability, respectively.

CYP2C9 polymorphism is also involved in the risk of bleeding during treatment with warfarin [52,53] or acenocoumarol [54,55,56], both at treatment initiation and maintenance.

In patients treated with warfarin the risk of bleeding increases from 1.77 in carriers of the 2C9*3 allele, to 1.91 for those with *CYP2C9*2* and 2.26 for those who have both alleles [41], while those treated with acenocoumarol have an increased risk of minor bleedings. [57]. Several studies have shown that *CYP2C9*2* has a minor effect on acenocoumarol dose [56,58], in contrast with the effect on warfarin. 

On the other hand, the increased sensibility of *CYP2C9 *3* carriers to acenocoumarol might be explained by the decrease of S-acenocoumarol metabolization [59]. This enantiomer—which normally doesn’t play any role in anticoagulation due to a short half-life of 1–2 h [59] will accumulate if its metabolization rate decreases, including in *CYP2C9*3* heterozygotes, thus prompting for a very low dose needed [59,60].

### 3.2. What Does VKORC1 Mean?

The *VKORC1* gene encodes the AVKs target enzyme—vitamin K epoxide reductase. Several *VKORC1* polymorphisms (SNP) have been recently identified in the gene-regulatory regions that are those mainly responsible for the warfarin dose variability and VKA resistance [61,62,63]. VKA resistance was defined as the need for very high doses of VKA to obtain an efficient anticoagulation—partial resistance, or the impossibility to obtain an INR in the therapeutic range—complete resistance, which is a rare phenomenon (1 in 1000 patients) [64].

Five of the *VKORC1* SNPs are most frequently found in Caucasians and are grouped together in two major haplotypes called A and B [14]. The studies of Reider et al [62] and Geisen et al. [65] on *VKORC1* haplotypes have analyzed several dozen enzyme polymorphisms, grouping them in haplotypes according to their effect on warfarin dose. The three major haplotypes (*VKORC1*2*: 42%, *VKORC1*3*: 38%, and *VKORC1*4*: 20%), identified by Geisen et al. [65] account for more than 99% of *VKORC1* genetic variability in Europeans and are closely correlated with inter-individual and interethnic variability of VKAs.

The review of Reider et al. has identified nine haplotypes, five of which have correlated to warfarin dose—H1 and H2 with the lowest dose while H7, H8, and H9 correlated with the highest dose.

*VKORC1*1* haplotype—considered the ancestral *VKORC1* haplotype—has a high prevalence in populations of African origin only, although it is not the most frequent haplotype in this population [66]. *VKORC1*2* (called “haplotype A” by Reider) encompasses H1 and H2 haplotypes, is associated with a high sensibility to warfarin and is found mostly in Asian populations, frequently in Caucasians and seldom in African populations [66]. *VKORC1*3* is the most frequent haplotype in the African population, followed by Caucasians (similar to *VKORC1*2*) and is the least frequent in Asian populations. *VKORC1*4* is similar to *VKORC1*3* but has the reference allele at *codon 9041* [66]. 

*VKORC1*3* și *VKORC1*4* are included in haplotype B that is associated with high warfarin dose. Minor differences aside, the H7 and H8 identified by Reider et al. [62] roughly correspond to the *VKORC1*3* group identified by Geisen et al. [65], while H9 by Reider et al. [62] (also from the B haplotype) correspond more closely to *VKORC1*4* identified by Geisen et al. [65]. Therefore, haplotype A polymorphisms (H1 and H2 or *VKORC1*2*) are associated with lower warfarin dose (2.9 and 3.0 mg/day), whereas haplotype B polymorphisms (H7, H8, and H9 or *VKORC1*3* and *VKORC1*4*) are associated with high warfarin dose (6.0 and 5.5 mg/day) [62].

Reider et al. [62] have shown that haplotype combinations cause a different expression of *VKORC1* and therefore are associated with different warfarin maintenance doses—2.7, 4.9 and 6.2 mg/day in people with haplotypes AA, AB and BB, respectively [63].

Of these, only *1639G>A* and *1173C>T* appear to be functionally related to VKA sensibility, both predicting the VKA dose in a similar fashion [63]

Therefore these two SNPs are found in most pharmacogenetic studies on VKAs, mostly SNP-*1639G>A* [61] that accounts for approximatively 20%–28% of dose variability in Caucasians but only 5%–7% of dose variability in African-Americans [61,67,68] mostly because of the lower prevalence of the *1639A* allele in this ethnic group [69].

Unlike *CYP2C9*, the *VKORC1* genotype doesn’t appear to impact the risk of bleeding with warfarin [70], but *VKORC1*−1693G>A polymorphism (17.6%) has a higher influence on acenocoumarol dose variability compared to *CYP2C9*2* and **3* polymorphisms (4.7%) [46,71].

Several recent studies on Caucasian and Asian populations have confirmed that *VKORC1 1639G>A, CYP2C9*2* and *CYP2C9*3* polymorphisms are the main genetic determinants of the warfarin dose in these populations [67,68]. Studies have shown that using a combination of gene mutations—*VKORC1-1639G>A* and *CYP2C9 (*2 și *3)* —and clinical factors (such as age, sex, weight, amiodarone use, etc.) could explain around 50%–60% [24,50] of the total variance of warfarin maintenance dose in Caucasians but only around 25% for African-Americans [69,72].

Several studies have found conclusive pieces of evidence on the increased ability of pharmacogenetic algorithms to predict the VKA dose and their usefulness in patients who need VKA treatment [73,74,75], both for warfarin [63,69,76,77,78,79] and acenocoumarol [46,49,80,81].

The U.S. Food and Drug Administration (FDA) has included the medical receipt of warfarin genotype-based dosing tables with recommended initial doses according to *CYP2C9* and *VKORC1-1639A* status [82], as pharmacogenetic algorithms have proven effective for establishing both the initial dose [51] and also the maintenance dose [83], even at 4–5 [84], or nine days after initiation [75]. Warfarin is used in several European countries (Italy, UK, Ireland etc.), USA and Canada, fluindione in France, phenprocoumon in Germany, Austria, Denmark, Switzerland, the Netherlands and Brazil, acenocoumarol in Spain, Germany, East European countries (the only VKA in Romanian pharmacopoeia) and Latin America [17]. Despite the fact that several studies have analyzed the impact of pharmacogenetics in prescribing the dose of acenocoumarol and have proposed several prescribing algorithms that have been validated on different populations (Dutch [47], Romanian [46], Spanish [49], French [81], Indian [80] and Greek [3]), at the moment there is no common consensus or universal algorithm available. An elegant review of the effectiveness of different dosing algorithms was conducted by Verhoef et al. The authors concluded that pharmacogenetics guided algorithms may predict the required coumarin dose before treatment initiation. The cost-effectiveness of genotype-guided coumarin dosing should be taken into account in comparison to NOAC [25]. Earlier studies indicated that using pharmacogenetic algorithms to guide VKA dosing resulted in a 10% absolute reduction in out-of-range INRs at one month, and a 66% lower rate of thromboembolic disease [85]. Later on, another study compared warfarin loading based on a pharmacogenetic algorithm comprising clinical variables with a loading regimen based on the clinical data alone. This study showed no improvement in either group in terms of the percentage of time in therapeutic range [86]. 2014 ESC Guidelines on the diagnosis and management of acute pulmonary embolism also noted, bases on trial evidence, that pharmacogenetic testing, used on top of clinical parameters did not show improvement of the anticoagulation quality [87]. Interestingly, in 2018 the largest meta-analysis (20 randomized controlled trials and 5980 adult patients) of genotype-guided vs. standard dosing of VKA was published by Kheiri et al [88]. The authors showed that the genotype-guided group had a better percentage of time in therapeutic range and a significant reduction in major and all bleeding events. However, there were no significant differences between the groups for over-anticoagulation (INR > 4), thromboembolism, other serious adverse events, or all-cause mortality [88]. In conclusion, such dosing algorithms should not be used on the general population in every-day activity, but in carefully selected cases who would benefit more from genetic analyses.

## 4. Practical Issues to Be Solved

### 4.1. How to Diagnose and Treat Patient in the Presence of the CYP2C9*2 Mutation (Heterozygote) and the VKORC C1173T Variant

Baring these in mind, the therapeutic choice and its surveillance of anticoagulant treatment could be difficult, such as the case of a 51-year old male, ex-smoker, with a recurring deep vein thrombosis (DVT) over the past 10 years. In the presence of congenital thrombophilia—factor V Leiden homozygote and protein S deficiency—chronic anticoagulant treatment with AVK was initiated. On acenocoumarol, INR was extremely variable (1.2 to 10 in the first 72 h of treatment), despite careful monitoring of dosage (acenocoumarol 0.93 mg/day) using the algorithm proposed by Pop et al. [26]. The clinical and demographic factors that might affect pharmacokinetics were ruled out, but for several months it was impossible to establish an adequate dose for efficient and safe anticoagulation. The patient had repetitive episodes of bleeding (epistaxis and painful calf muscles) and variable INR. After an episode of severe, low gastrointestinal bleeding, genetic testing for sensitivity/resistance to AVK was performed. The patient was found positive for the *CYP2C9*2* mutation (heterozygote) and also for the *VKORC-C1173T* variant (heterozygote).

There are two important issues to discuss in this particular case: what is the significance of the association of these two mutations in this patient’s treatment and prognosis, and how do these mutations influence the bridging phase during the initiation of the VKA therapy.

First of all, regarding the significance of the association between these two mutations, we know for now, that patients who possess one or more genetic variations in *CYP2C9* and *VKORC1* are at risk for adverse drug events with VKA and require significant dose reductions to achieve therapeutic INR [89]. The identified *CYP2C9*2* polymorphism has been described in other studies as a major factor affecting warfarin sensitivity, so patients with this polymorphism should start anticoagulant treatment with acenocoumarol instead of warfarin to avoid the risk of over-anticoagulation and subsequent bleeding [46]. But in our case, the acenocoumarol treatment did not provide an effective and constant INR because the *VKORC C1173T* decreases *VKORC1* activity leading to a marked inhibition of VKAs metabolism, resulting in a much longer elimination half-life of the drug (warfarin or acenocoumarol) with an increased risk of bleeding complications, and unstable INR even at lower doses [90].

This aspect was also pointed out by Shuen et al. [64], who found that 1.6% of their patient group, although carriers of the *VKORC1-1639G>A* allele (that increases warfarin sensitivity), actually required increased doses. Under these circumstances, certain ethnic groups might be disadvantaged by the use of algorithms containing only the most frequent mutations, such as *CYP2C9*2* and **3*, *1639G>A* and *1173C>T VKORC1* polymorphisms.

Secondly, we consider it worthwhile to mention how the initial treatment with VKA is affected by warfarin genetics and how bridge therapy should be applied in this type of patient.

Most of the studies regarding gene mutation effect on therapy focused on the initial phase of therapy when the needed dose of warfarin is not yet known and is titrated according to the INR. Moreover, all the guidelines and algorithms of prediction focused on initial dosing, even though major bleeding events occurred in the later phase of the treatment [91,92]. Schwarz et al. showed that “the *CYP2C9* genotype and *VKORC1* haplotype had a significant influence on the required warfarin dose after the first 2 weeks of therapy” [91]. Ferder al. found that *CYP2C9* and *VKORC1* genotypes “were significant independent predictors of therapeutic dose at each weekly interval”, and the capacity of their predictive ability decreased in time. After the first week of treatment INR and prior were more predictive and the genotype less relevant for the optimal therapeutic dosing [92]. On the other hand, Kawai et al. highlighted in their study that genotype information has a predictive ability beyond the first month of treatment and that *CYP2C9*3* affects the risk of bleeding, even one month after warfarin initial dose [83]. Mutations in *VKORC1* and *CYP2C9* increase the risk of over-dosing during the first 30 days of therapy before the dose for optimal INR range was obtained [93]. Most of the studies focused on Warfarin and not on acenocoumarol, so most of the information is extrapolated to acenocumarol.

In this particular case with two genetic mutations affecting VKA metabolism, using the prediction algorithm to obtain a certain AVK dose (i.e. a dosing algorithm) would not have been adequate, neither during the bridging phase nor later. On the other hand, using a diagnosis algorithm to identify both mutations was beneficial for the overall outcome.

This report summarizes the clinical course of a supratherapeutic response to acenocoumarol in a 51-year old male white male due to impaired VKA metabolism as a result of a *CYP2C9*2* mutation genotype as well as increased pharmacodynamic sensitivity due to the presence of *VKORC C1173T* variant. Thus, given the difficulties in obtaining an INR in the therapeutic range and the association of major bleeding, we considered appropriate to switch on NOAC. We chose Dabigatran which is metabolized independently of cytochrome P450 enzymes or on *VKORC1* and the switch proved to be beneficial for our patient. This case proves that the absence of *CYP2C9* and/or *VKORC1* pharmacogenetic testing in clinical practice is both challenging and risky for patients and physicians alike. Moreover, it underlines that NOAC could be a saving solution for this rare category of patients and its administration in such cases should be more extensively studied.

### 4.2. Genetic Testing—How Should We Do It, How Far Should We Go?

There is a case of a previously healthy 41-year-old man being diagnosed in the ER with popliteal DVT and treated with LMWH anticoagulation (Enoxaparin) and VKA (acenocoumarol) according to the local protocol (a loading dose of 4 mg on the first and second day and 2 mg per day on the third and fourth day). The patient chose VKA over NOAC treatment due to financial reasons. On the fifth day, after the patient received a total dose of 12 mg acenocoumarol, INR was 1.14. As a result, the dose was raised to 6 mg per day for the next two days, however, when retested on day seven, INR was 1.13. We ruled out other possible causes for acenocoumarol resistance (non-compliance with the treatment plan, other concomitant medications, dietary factors, etc.) [94] and we suspected that treatment failure was most likely the result of partial or total resistance to coumarin.

We, therefore, performed genetic testing that showed a *VKORC1 ASP 36 Tyr* polymorphism, a marker specifically associated with the highest needed dose (over 70 mg per week) to maintain an effective INR.

Finding evidence of the genetic determinism of acenocoumarol resistance mandates a careful consideration of the impact on the pharmacokinetics and pharmacodynamics of the prospective medication. One can choose either to carefully titrate the VKA dose (warfarin or acenocoumarol) with the risk of major complications as we’ve seen in the previous case or to go for a NOAC. In this case, we chose to treat the patient with a NOAC (Dabigatran), for an initial period of 3 months, subsequently extended to 6 months, when the complete resorption of the thrombus was achieved. During the next 12 months after discontinuation of the anticoagulant treatment, the patient showed no clinical or ultrasound evidence of a new DVT.

In this case, pharmacogenetic algorithms that include classical clinical and demographic factors and additional common markers such as *VKORC1 c.-1639G>A*, *CYP2C9*2* and **3* could not predict the dose of VKAs. Therefore, although the frequency of *VKORC1 Asp36Tyr* mutation is low (1.6% in the heterogeneous population) we also support the assertion by Shuen et al include it in VKAs dose prediction algorithms, especially in Caucasians. *Asp36Tyr* polymorphism has a “dominant” effect over the warfarin-reducing effects of *VKORC1 1639G>A* [95,96]. Studies have found 26 mutations of *VKORC1* gene that are associated with increased warfarin dose [37]. Of these, *Asp36Tyr* and *Val 66Met*, have increased frequency in certain geographical areas such as Israel, Ethiopia, Russia, Africa and Spain and appear to be of greater relevance, while *Arg12Arg* and *Leu120Leu* (two polymorphisms of the silent coding region) have been identified by Anton et al. [97] in Spaniards who required high doses of acenocoumarol. Other studies have shown the presence of 13 mutations in *VKROC1*, including *Asp36Tyr* [37] and *L128R* [97] in patients requiring increased doses of warfarin. Moreover, a number of studies have shown that the *Asp36Tyr* mutation of *VKORC1* has an ethnic distribution, being widespread in the Jewish population of Ethiopian origin (allele frequency 15%), in that of Northeastern African and Middle-Eastern populations, and less frequent in Ashkenazi Jews (4%) or in Sephardi Jews (0.6%) [65]. Genotype *L128R* has been described only in rare cases of warfarin resistance [37,95].

Shuen et al argued that adding the *VKORC1 p.Asp36Tyr* to the model with *VKORC1* and *CYP2C9* would increase its explanatory/predictive capacity by 10.7%, or about 50% of the dose variability in Caucasians [64]. Data show that the screening for *Asp36Tyr* mutation even in populations where its frequency is low [75] improves anticoagulant treatment efficiency.

This case illustrates that including *Asp36Tyr* mutation in the diagnosis algorithm for identifying the genetic cause of INR variability could improve both the diagnosis and management of particular patients with VKA resistance. Therefore, we propose such an algorithm comprising the above-mentioned polymorphism, as in Figure 2 [98]. When reaching efficient anticoagulation with VKA becomes difficult, after excluding demographic factors that might affect the lability of the INR, one should focus on genetic testing.

First of all, if INR values are above 3.5, we consider that patients should be tested for polymorphisms associated with sensitivity to VKA, namely *VKORC1 1639G>A, CYP2C9*2 and *3* (the most prevalent in Caucasian population according to Shuen et al [1]. If these mutations are validated, there are two options to take into account: either integrate the clinical and genetic factors in a validated pharmacogenetic algorithm for Acenocumarol [46,48] and calculate the predicted dose of VKA for reaching therapeutic anticoagulation or switch directly to NOAC. But if the genetic tests are negative, an option could be to further test for the presence of *CYP4F2* mutations, as it was previously mentioned that it is also linked to VKA sensibility. A positive result will determine two alternatives: switch directly to NOAC or integrate the mutant variant in a pharmacogenetic algorithm like the one proposed by [49]. However, if these results are negative, the third panel of mutations to be considered are those related to *CYP2C9*5, *6, *8 or *11*. The presence of mutant variants implies integrating it in an algorithm like the one proposed by Gauge et al. for Warfarin [99] and calculate the required dose accordingly, or switch to NOAC. If there is a negative response, one should stop genetic testing and treat the patient with NOAC if affordable.

Secondly, if the INR values are below 2 with a proper loading regimen after eliminating the environmental and pharmacokinetic factors, one should have in mind testing for polymorphisms associated with coumarin resistance, such as *VKORC1 ARG36 Tyr* and *VKORC1 Val66 Met*, being the most prevalent as mentioned above. Furthermore, if genetic testing proves to be positive for either one of them, a feasible solution is to switch to NOAC. However, when the results are negative for these mutations, one should focus on finding other mutations associated with resistance prevalent in the population of interest. If this approach is not possible, a universal solution is to treat the patient with NOAC if no contraindications exist. Such an algorithm as the proposed could be useful for uncontrolled patients but it should be very carefully evaluated in larger studies, before widespread use.

## 5. NOAC the Practical Solution

Warfarin has been superseded in most areas now by the use of NOAC, and large studies and metanalysis have pointed to the improved overall safety of NOAC compared to warfarin. The meta-analysis of Ruff et al. comprises the four most important randomized clinical trials of NOAC vs. warfarin in patients with non-valvular atrial fibrillation. The authors showed that NOAC were associated with a significant 19% relative risk reduction (RRR) in any stroke or systemic embolism significant 10% RRR in all-cause mortality and a 14% RRR in major bleeding (*p* = 0.06). Moreover, all NOACs consistently reduced haemorrhagic stroke or any intracranial bleeding for > 50% (both *p* < 0.0001) [100]. Recently, Ntaios et al. summarized in an elegant analysis the landmark studies proving the effectiveness and safety of NOAC relative to warfarin [101]. In 2018, Hohnloser et al. reported the first retrospective observational comparison of the effectiveness and safety of NOAC vs. phenprocoumon in a large atrial fibrillation cohort (61,205 patients), and their findings were broadly confirming the results of NOAC vs. warfarin landmark trials [102]. Some of these data regarding NOAC were also summarized by Joppa et al. [103]

Considering their pharmacokinetic and pharmacogenetics, we believe that NOAC could be a valuable treatment option for these rare types of patients, initially having a clear indication for VKA. Dabigatran (a direct thrombin inhibitor) and factor Xa inhibitors rivaroxaban, apixaban and edoxaban (not approved in all the European countries), are new oral anticoagulants approved for the treatment of atrial fibrillation and thromboembolic events [104].

Dabigatran is administered as a pro-drug that is converted by the liver esterase *CES1* to the active drug [105,106,107]. Maximum plasma concentrations of dabigatran are achieved at 1–3 h after intake [107] and is excreted predominantly by the kidneys (80%) [83]. Dabigatran etexilate is a substrate of the P-glycoprotein intestinal efflux transporter (an efflux pump for xenobiotics) encoded by the *ABCB1* gene and does not inhibit cytochrome P450 (*P*); therefore, its potential for drug-drug interactions is low [107]. On the other hand, P-glycoprotein inhibition and renal failure are two major independent factors that can increase dabigatran concentrations, with greater effects if both are present. Thus, strong P-glycoprotein inhibitors increase the effect of dabigatran as follows: dronedarone (by 73% to 99%), amiodarone (by 50% to 58%), quinidine (by 53% to 56%), and ketoconazole (by up to 153%). Other P-glycoprotein inhibitors such as clarithromycin led to a two-fold increase in Dabigatran concentration irrespective of the *ABCB1* genotype [108]. A reduced dose of dabigatran is recommended when another P-glycoprotein inhibitor is administered [109]. The active metabolite of dabigatran presented a higher inter-individual variability in blood concentrations. Allele variants of ABCB1 and of *CES1* may play a pivotal role in this inter-individual variability. Dimatteo et al. showed that *CES1 SNP rs8192935* may play a significant role in modulating dabigatran trough concentrations. The authors proposed that screening of this polymorphism might be useful to identify patients at risk for adverse events and needing different intensity regimens of anticoagulation [110].

Rivaroxaban oral bioavailability was reported to be over 80% and achieves maximal anticoagulation 2–4 h after administration [111]. Excretion of rivaroxaban occurs through two main pathways: cytochrome P450 *CYP2J2* and *CYP3A4*-dependent metabolism are responsible for two-thirds of its elimination while one-third is excreted unchanged by the kidneys [112]. There has been inter-individual variation in exposure and response to rivaroxaban. Other P-glycoprotein inhibitors, including and ritonavir severely reduced the drug efflux and thus such co-therapy should be administered with caution while P-glycoprotein inducers (rifampin) should be avoided because it increases the risk for stroke [113]. When erythromycin, clarithromycin, and fluconazole were administered, rivaroxaban plasmatic levels increased by 34%–54%, whereas coadministration of ketoconazole and ritonavir led to an increase of 153%–158%. Food components, dietary supplements, and other over the counter drugs affect the P-glycoprotein system and the efficiency of anticoagulation [114].

Apixaban, like rivaroxaban, is a direct oral inhibitor of the Xa factor, being also a substrate for the P-glycoprotein transporter but mainly for *CYP3A4/5* [115] and also *CYP1A2, 2C8, 2C9, 2C19, 2J2* [116]. Apixaban has also been reported as a substrate for P-glycoprotein [117]. Animal experiments suggested that apixaban has low penetration through the blood-brain barrier. Again, the efflux transporters P-glycoprotein are assumed to prevent or reduce drug entry [104].

Taking into account the proteins associated with the metabolism of rivaroxaban and apixaban it is possible that polymorphisms in the *CYP* genes (*CYP3A4/5, CYP 1A2, 2C8, 2C9, 2C19, and 2J2*) could be associated with variability of anticoagulant effect, similar to the results found with warfarin and acenocoumarol [108]. All these considered, we chose dabigatran for the two cases presented above.

## 6. Conclusions

In conclusion, despite the new knowledge about the genetic determinism of the variability of the efficacy of anticoagulant treatment and the use of pharmacogenetic algorithms, its management still remains a challenge. Therefore, genetic testing (although rarely performed in clinical practice before initiating anticoagulant therapy [76], due to time and availability constraints), could become extremely important, especially in particular situations (with increased sensitivity or VKAs resistance), both for increasing efficacy and safety. At the same time, due to their different pharmacogenetics, NOAC could represent a worthy treatment alternative in these situations.

## Figures and Tables

**Figure 1 jcm-08-01747-f001:**
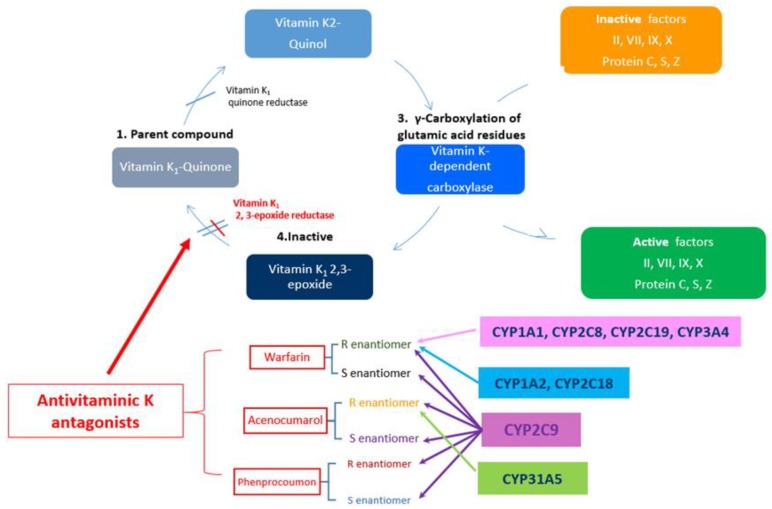
Mechanism of action of VKA. *CYP1A1* Cytochrome P450 family 1, subfamily A, polypeptide1; *CYP1A2* Cytochrome P450 family 1, subfamily A, polypeptide 2; *CYP2C18* Cytochrome P450 family 2, subfamily C, polypeptide 18; *CYP2C19* Cytochrome P450 family 2, subfamily C, polypeptide19; *CYP3A4* Cytochrome P450 family 3, subfamily A, polypeptide 4; *CYP2C9* Cytochrome P450 family 2, subfamily C, polypeptide 9; *CYP2C3A5* Cytochrome P450 family 3, subfamily A, polypeptide 5. Adapted after [38,39].

**Figure 2 jcm-08-01747-f002:**
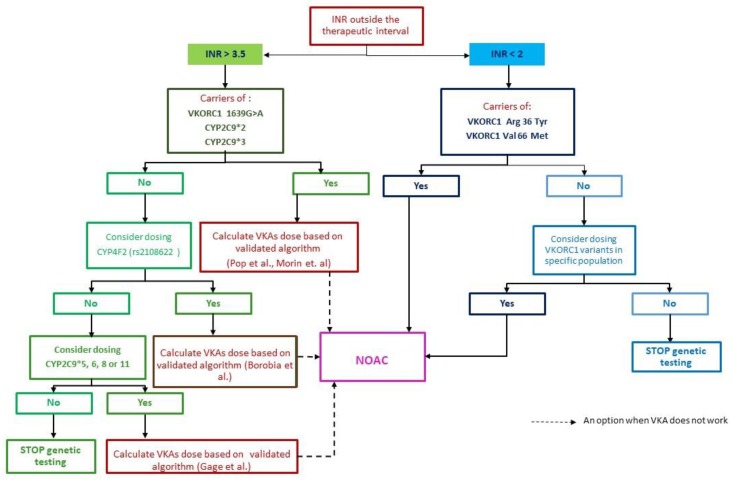
Proposed algorithm for identifying the genetic cause of INR variability after excluding environmental factors. Adapted after [98].

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
