# Peer review of "Oral Anticoagulant Therapy—When Art Meets Science"

_jcm, 2019, doi:10.3390/jcm8101747_

Round 1

Reviewer 1 Report

Thank you for extensive review on warfarin genetics

The authors hint to the differential bleeding risk in different treatment phases. It would be worthwhile to devote a section to the first 7 days of warfarin treatment and how this phase is affected by warfarin genetics as well as to the bridging phase and make reference to the bridge study since this study did not account for variability in warfarin kinetics - is should patienbts with certain phenoytpes bridge or not - is the risk differential on the genetics ?

Overall however warfarin has been superseded in most areas now by the use of DOAC and your review should point to the Ruff Metanalysis regarding the improved overall safety of DOAC compared to warfarin. Your review is incomplete without delineating where warfarin is now still recommended and it would be nice here to make special mention of APS and mechanical heart valves.

2 factors that I have personally encountered is congenital factor VII deficiency and acquired factor VII inhibitors which affect warfarin control and maybe entirely unexpected so important to include in the differential

Also a few lines on the history of warfarin in the introduction would be nice to round off the paper.

Author Response

Dear Editor,

Thank you for giving us the opportunity to submit a revised draft of our manuscript entitled “Oral anticoagulant therapy – when art meets science” to Journal of Clinical Medicine. We appreciate the time and effort that you and the reviewers have dedicated to providing your valuable feedback on our manuscript. We are grateful to the reviewers for their insightful comments on the paper. We have been able to incorporate changes to reflect most of the suggestions provided by the reviewers. All the changes within the manuscript are highlighted in red.

Here is a point-by-point response to the reviewers’ comments and concerns.

Review 1

 Thank you for extensive review on warfarin genetics

 R1. The authors hint to the differential bleeding risk in different treatment phases. It would be worthwhile to devote a section to the first 7 days of warfarin treatment and how this phase is affected by warfarin genetics as well as to the bridging phase and make reference to the bridge study since this study did not account for variability in warfarin kinetics - is should patients with certain phenotypes bridge or not - is the risk differential on the genetics? 

A1. A paragraph about initial phase monitoring and genotype was introduced along with references chapter 4.1 paragraph 293-309

R2. Overall however warfarin has been superseded in most areas now by the use of DOAC and your review should point to the Ruff Metanalysis regarding the improved overall safety of DOAC compared to warfarin. Your review is incomplete without delineating where warfarin is now still recommended and it would be nice here to make special mention of APS and mechanical heart valves.

A2. The overall safety and efficiency of NOAC was discussed in chapter 5 paragraph 293-309 with refernces including Ruff et al. Indications of VKA were introduced with special mention about APS and thrombophilia in chapter 2 paragraph 47-68

R3. 2 factors that I have personally encountered is congenital factor VII deficiency and acquired factor VII inhibitors which affect warfarin control and maybe entirely unexpected so important to include in the differential diagnosis.

A3. A paragraph was introduced in Chapter 2 paragraph 106-115 about the importance of congenital factor VII deficiency and acquired factor VII inhibitors

R4. Also a few lines on the history of warfarin in the introduction would be nice to round off the paper.

A4. A paragraph was introduced about the history of Warfarin in chapter 2 paragraph 69-80.

Reviewer 2 Report

In general, I find this review not very clearly focused.

Authors do not mention and review the evidences oon the utility of the different algorithms in dosing warfarin or acenocoumarol, as there are a significant number of clinical trials evaluating them, with no consistent results.

Writing is not very clear or contain errors in several parts of the manuscript:

Lines 40 to 44 - paragrphs in lines 314 to 320 and 322 to 328 are the same.

In my opinion, the data on use of DOAC vs coumarins are quite old considering the rapid increase of market share of DOAC, at least in developed countries. I would suggest authors to look for more recent data.

As far as I know, the most frequently used coumarin in Spain is acenocumarol. Authors should check the data provided in lines 193 to 194.

Lines 223-225: It is no clear to me how the dose of acenocoumarol has been calculated, as the reference given (47) related only to warfarin. Even, authors should clarify how they calculated the dose for warfarin.

I do not understand the proposed algorithm. I feel that it requires more extensive explanation. Also, a proposal of this type should provide some validation of its utility.

The cases reported, especially the first one, are incomplete and do not provide clear insight in the underlying mechanisms.

In my opinion the last part of the review (Genetic determinism of the response to novel oral anticoagulants) do not “match” the global manuscript and provide very scarce information. In addition, authors do not provide clear information about the existence of relevant variability in DOAC associated to pharmacogenomics variability.

Author Response

We appreciate the time and effort that you have dedicated to providing your valuable feedback on our manuscript. We are grateful to you for the insightful comments on the paper. We have been able to incorporate changes to reflect most of the suggestions you provided. All the changes within the manuscript are highlighted in red.

Here is a point-by-point response to your comments and concerns.

Review 2

R1. In general, I find this review not very clearly focused.

A1. The objective of the review was introduced in the abstract and in Chapter 2 paragraphs 121-124

R2. Authors do not mention and review the evidences on the utility of the different algorithms in dosing warfarin or acenocoumarol, as there are a significant number of clinical trials evaluating them, with no consistent results.

A2. Data regarding different algorithms efficiency were introduced in Chapter 3 paragraphs 153-157

 R3. Writing is not very clear or contain errors in several parts of the manuscript:

A3. Lines 40 to 44 (in the previous ante-review document) were modified – lines 41-43 in this document. Paragraphs in lines 314 to 320 and 322 to 328 are the same, were modified.

R4. In my opinion, the data on use of DOAC vs coumarins are quite old considering the rapid increase of market share of DOAC, at least in developed countries. I would suggest authors to look for more recent data.

A4. The exact indications of VKA were introduced in chapter 2 line 47-68. Recent data regarding the safety and effectiveness of NOAC were introduced in Chapter 5

R5. As far as I know, the most frequently used coumarin in Spain is acenocumarol. Authors should check the data provided in lines 193 to 194.

A5. Paragraph 193-194 was reformulated – paragraphs 246-249 in the present document

R6. Lines 223-225: It is no clear to me how the dose of acenocoumarol has been calculated, as the reference given (47) related only to warfarin. Even, authors should clarify how they calculated the dose for warfarin.

A6. The reference 47 was replaced with 43. The dose of acenocumarol was calculated using the specific algorithm developed earlier on Romanian population (Reference 43 Pop et al). The paragraph about Warfarin was excluded. The tested algorithms for VKA dosing were included in Figure 2.

R7. I do not understand the proposed algorithm. I feel that it requires more extensive explanation. Also, a proposal of this type should provide some validation of its utility.

A7. The algorithm was modified. Explanation of the algorithm was added chapter 4 lines 378-400. The algorithm was used in the presented cases but the authors confirm that more extensive studies are needed to validate its utility.

R8. The cases reported, especially the first one, are incomplete and do not provide clear insight in the underlying mechanisms.

A8. Both cases were rewritten.

R9. In my opinion the last part of the review (Genetic determinism of the response to novel oral anticoagulants) do not “match” the global manuscript and provide very scarce information. In addition, authors do not provide clear information about the existence of relevant variability in DOAC associated to pharmacogenomics variability.

A9. Clear data regarding NOAC pharmacogenetics was provided and the title of the chapter was changed

Round 2

Reviewer 2 Report

The manuscript is clearer and best focused. However, I still feel that a brief mention to the clinical trials (and metaanalysis) evaluating the utility of pharmacogenetic algorithms in dosing of warfarin and acenocoumarol should be done. These clinical trials have non fully conclusive results and, in fact, point to the need to re-think the use of pharmacogenetic information in VKA dosing, due to the complexity of this issue. An algorithm as the proposed could be useful for uncontrolled patients but it should be very carefully evaluated before widespread use. 

Author Response

Reviewer 2

The manuscript is clearer and best focused.

R1. However, I still feel that a brief mention to the clinical trials (and metaanalysis) evaluating the utility of pharmacogenetic algorithms in dosing of warfarin and acenocoumarol should be done. These clinical trials have not fully conclusive results and, in fact, point to the need to re-think the use of pharmacogenetic information in VKA dosing, due to the complexity of this issue.

A1. A paragraph mentioning the clinical trials and metanalysis about genetic algorithms was introduced (lines 256-271) along with references (80-83).

R2. An algorithm as the proposed could be useful for uncontrolled patients but it should be very carefully evaluated before widespread use. 

A2. A paragraph was introduced in lines 413-415